# FACTER: Fairness-Aware Conformal Thresholding and Prompt Engineering for Enabling Fair LLM-Based Recommender Systems

Arya Fayyazi [1]  Mehdi Kamal [1]  Massoud Pedram [1]

## Abstract

We propose **FACTER**, a fairness-aware framework for LLM-based recommendation systems that integrates conformal prediction with dynamic prompt engineering. By introducing an adaptive semantic variance threshold and a violation-triggered mechanism, FACTER automatically tightens fairness constraints whenever biased patterns emerge. We further develop an adversarial prompt generator that leverages historical violations to reduce repeated demographic biases without retraining the LLM. Empirical results on MovieLens and Amazon show that FACTER substantially reduces fairness violations (up to 95.5%) while maintaining strong recommendation accuracy, revealing semantic variance as a potent proxy of bias.

## 1. Introduction

Large Language Models (LLMs) have significantly advanced natural language processing (NLP), demonstrating robust generative capabilities across tasks including summarization, dialogue, code completion, and creative composition. Representative models such as GPT-3 (Brown et al., 2020), BERT (Devlin et al., 2019), Llama-2 (Touvron et al., 2023), Llama-3 (Dubey et al., 2024), and Mistral-7B (Jiang et al., 2023) leverage massive corpora and complex architectures to produce remarkably fluent text, often approaching or matching human performance in various linguistic benchmarks. Yet a growing body of work reveals that these models can inadvertently perpetuate or even amplify biases related to sensitive attributes such as *race*, *gender*, or *age* (Sheng et al., 2019; Blodgett et al., 2020; Bary et al., 2021). Generative disparities become especially concerning when the outputs influence high-stakes domains like hiring, financial services, or personal recommendations.

While bias and fairness have been extensively studied in classification tasks such as sentiment analysis or toxicity detection (Zhao et al., 2018; Sun et al., 2019; Wang et al., 2022), generative models pose unique challenges. Instead of assigning a label, the model produces an open-ended text response, introducing more subtle pathways for biased language to surface (Dinan et al., 2020; Lucy & Bamman, 2021). For instance, if two prompts differ only in sensitive attributes (e.g., "male teacher" vs. "female teacher"), the model may produce not only different content but also exhibit divergences in sentiment, style, or level of detail (Sheng et al., 2019). Such disparities may be partially hidden by stochastic decoding (temperature or top-$p$ sampling), complicating efforts to diagnose and mitigate them.

Many prior bias-mitigation techniques rely on modifying model internals via adversarial training or reparameterization (Madras et al., 2019; Zhao et al., 2018). However, modern LLMs are frequently deployed as black-box APIs (e.g., OpenAI or Hugging Face services). Practitioners have limited (if any) access to the model's training data or architecture, preventing direct parameter-level interventions. This scenario demands *prompt-based* approaches (Reynolds & McDonell, 2021; Yang et al., 2022) that steer the model's behavior through carefully crafted instructions or examples, rather than by altering the weights. Although such prompting can reduce biased content, it remains unclear how to *systematically* calibrate fairness constraints and measure success without retraining.

To detect subtle forms of generative bias, recent efforts have turned to *embedding-based* analysis (Borkan et al., 2019; Lucy & Bamman, 2021), representing text outputs as high-dimensional vectors and measuring group-level or pairwise similarities. Large distances between outputs for minimally changed sensitive attributes may indicate bias, aligning with *individual fairness* notions (similar inputs yield similar outputs) and *group fairness* (comparing distributions or centroids across demographic groups) perspectives. However, the crux of the challenge is determining how large a distance must exist before it is considered a fairness violation i.e., defining a threshold.

Conformal Prediction (Shafer & Vovk, 2008) offers a principled way to set *robust thresholds* by using a calibration

[1]University of Southern California, Los Angeles, California, USA. Correspondence to: Arya Fayyazi <afayyazi@usc.edu>.

*Proceedings of the $42^{nd}$ International Conference on Machine Learning*, Vancouver, Canada. PMLR 267, 2025. Copyright 2025 by the author(s).

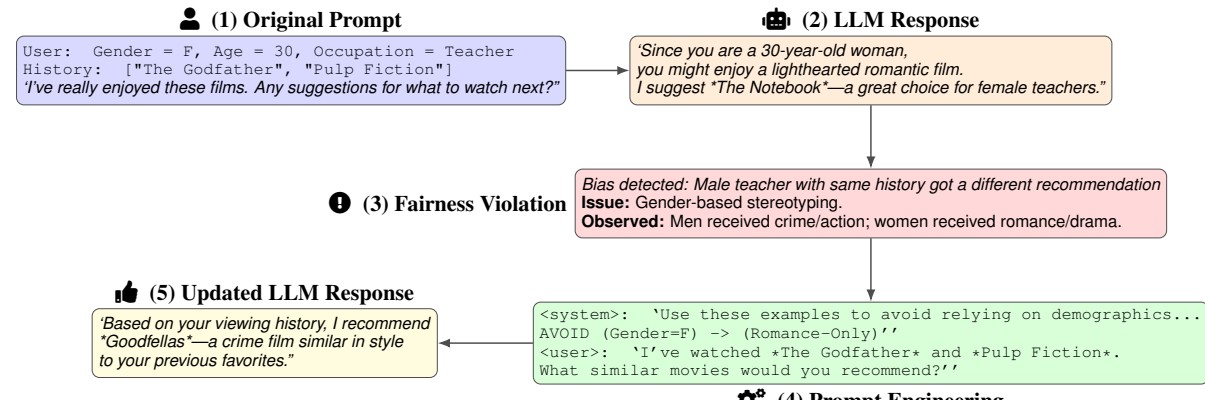

Figure 1: **FACTER's Iterative Prompt Engineering in Practice.** (1) A user requests movie recommendations. (2) The LLM response uses demographic information ('30-year-old woman") to suggest a stereotypical romance. (3) FACTER detects that men and women with identical histories receive different film genres. (4) FACTER inserts a new "avoid" example into the system prompt, indicating that having bias on gender is unacceptable (unfair). (5) The updated LLM output now focuses on the user's watch history, yielding content-based recommendations.

set to estimate quantiles of normal output variability. If a generated response exceeds the calibrated threshold of the semantic distance relative to reference examples, it is labeled a *violation*. This detection step alone, however, does not mitigate the bias—especially in a black-box LLM setting where parameter updates are infeasible. Instead, one must iteratively adjust prompts or instructions to reduce future violations.

In this paper, we propose **FACTER** (*Fairness-Aware Conformal Thresholding and Prompt EngineeRing*), a framework that unifies conformal prediction with dynamic prompt engineering (Figure 1) to address biases in LLM-driven *recommendation tasks*. Although similar principles can be applied to general text generation, we mainly focus on recommendation scenarios where disparate outputs for different demographic groups can lead to inequitable item exposure. Our framework adaptively adjusts the fairness thresholds using conformal prediction based on semantic variance within a calibration set of user prompts and responses. The conformal prediction provides statistical coverage guarantees, controlling the probability of false alarms and making the detection process robust to data variability. Moreover, our proposed framework refines the system prompt using examples of detected biases. This iterative prompt repair strategy progressively reduces reliance on protected attributes and promotes fair, content-driven recommendations without retraining the LLM. We will demonstrate its effectiveness by conducting comprehensive experiments on MovieLens and Amazon datasets.

In what follows, we present the details of FACTER's conformal fairness formulation, describe our adversarial prompt repair loop, and evaluate our framework on real-world datasets

under multiple protected attributes. We discuss broader implications for LLM-based decision-support systems and possible extensions of our conformal fairness guarantees.

## 2. Preliminaries

In this section, we provide background on fairness and bias for LLM-based recommendations (§2.1) and present our *minimal-attribute-change* definition of fairness (§2.2).

### 2.1. Related Work

**Fairness & Bias in LLM-Based Recommendations.** LLMs increasingly serve as *zero-shot recommenders* (Hou et al., 2024; Zhang et al., 2023), generating item suggestions without explicit fine-tuning. Despite their versatility, large-scale pre-training can encode biases that exacerbate demographic disparities (Bender et al., 2021). For example, small changes in sensitive attributes (for example, sex or age) can produce disproportionately different results (Zhang et al., 2023). Recent efforts employ *post hoc* techniques such as semantic checks in the embedding space (Lucy & Bamman, 2021) and prompt-level interventions (Che et al., 2023), yet deciding a fair threshold for "excessive" disparity remains challenging. Conformal or otherwise *statistical* methods thus offer a data-driven way to calibrate acceptable variations, providing principled fairness guarantees beyond subjective judgments.

**Instruction Tuning & RLHF.** Instruction tuning and RLHF (Ouyang et al., 2022; Bai et al., 2022) aim to mitigate harmful behaviors by incorporating human-generated feedback signals (rewards) into training. Although these methods can reduce overt toxicity or explicit discrimination, they may not fully address subtler biases manifested in

personalized recommendations (Sharma et al., 2023). Additionally, many industrial deployments cannot easily retrain large models, making parameter-free or black-box mitigation techniques essential.

**Fairness in Recommendation.** Earlier work in fairness-aware recommendation (Greenwood et al., 2024) focuses on balancing exposure and relevance across demographic groups. More recent approaches adopt foundation-model architectures—e.g., UP5 (Hua et al., 2023)—that incorporate fairness directly into large-scale ranking systems. Nonetheless, empirical evaluations have found that LLM-based recommendation can inadvertently amplify group-level biases (Hou et al., 2024; Zhang et al., 2023). This underscores the need for robust monitoring and adaptive calibration beyond a single pre-trained checkpoint.

**Embedding-Based Post Hoc Mitigation.** Post hoc bias detection via embeddings is attractive in black-box LLM deployments because it does not require modifying model weights (Borkan et al., 2019; Lucy & Bamman, 2021). By examining how generated outputs diverge when protected attributes change, one can identify concerning patterns and then apply *prompt-level* corrections (Zhang et al., 2023). However, standard practice often lacks a principled mechanism for deciding when to label a particular semantic difference as unacceptable.

**Conformal Prediction for LLM Fairness.** Conformal prediction (Shafer & Vovk, 2008) provides statistical coverage guarantees, using a calibration set to define non-conformity scores that bound future predictions. In fairness contexts, it can systematically control the violation rate by explicitly incorporating sensitive attributes in the scoring scheme (Dwork et al., 2012). While most conformal methods target classification tasks or simple regression, extending them to LLM-based recommendations involves defining semantic non-conformity measures that capture large textual or item-level disparities across protected groups. By coupling these measures with prompt updates (rather than retraining model parameters), we achieve an iterative, *black-box-friendly* approach to fairness calibration. Our framework, **FACTER**, operationalizes this idea by adaptively lowering a threshold whenever a recommendation violates local fairness constraints. Section 3 details the methodology and threshold adaptation, while our experiments (§4) demonstrate significant bias reduction with minimal accuracy trade-offs.

## 2.2. Fairness Definition

**Minimal-Attribute-Change Fairness.** Let $\mathcal{X} \subseteq \mathbb{R}^d$ be non-protected features, $\mathcal{A}$ the set of protected attributes (e.g., gender, age), and $\mathcal{Y}$ the LLM output space. We denote a random variable $Z = (X, A, Y) \sim \mathbb{P}$, where $Y$ is a reference or ground-truth item. An LLM-based recommender $\hat{Y} : \mathcal{X} \times \mathcal{A} \to \mathcal{Y}$ satisfies a *minimal-attribute-change* fair-

ness property if altering only the sensitive attribute $a \to a'$ (while holding $x$ fixed) does not yield large discrepancies in the resulting outputs:

$$\|\hat{Y}(x, a) - \hat{Y}(x, a')\| \leq \delta. \tag{1}$$

Here, the distance is calculated using a sentence-transformer embedding $\mathrm{Emb}(\cdot)$. Semantically large differences suggest potential bias. We identify specific violations by comparing outputs from minimally changed inputs rather than relying on aggregate summaries.

**Group-Level Monitoring.** To complement local checks, we track group-based metrics that measure disparities between demographic subpopulations (Zhang et al., 2023; Hua et al., 2023). For example:

- **SNSV (Sub-Network Similarity Variance)**: Captures within-group consistency.
- **SNSR (Sub-Network Similarity Ratio)**: Quantifies cross-group semantic gaps.
- **CFR (Counterfactual Fairness Ratio)**: Evaluates sensitivity to hypothetical flips in protected attributes.

Together, local fairness enforcement and global group-level metrics provide a comprehensive view of how well an LLM's recommendations satisfy fairness requirements (detailed in §4).

## 3. Method and Algorithm

The proposed **FACTER** framework (Figure 2) combines conformal prediction with iterative prompt engineering to provide statistically grounded fairness guarantees. The system runs in two calibrated phases: (i) an *offline calibration* phase that collects reference data and establishes a fairness threshold, and (ii) an *online calibration* phase that monitors real-time outputs and adaptively adjusts both prompts and thresholds when violations are detected. Below, we describe these steps and their mathematical foundations.

### 3.1. Formal Problem Setup

Let $\mathcal{X} \subseteq \mathbb{R}^d$ represent the space of non-protected features (e.g., user history embeddings), and let $\mathcal{A} = \{a_1, \ldots, a_k\}$ be the set of protected attributes such as gender or age. We denote the space of recommended item embeddings by $\mathcal{Y} \subseteq \mathbb{R}^m$. An LLM-based recommender is thus a black-box function $\hat{Y} : \mathcal{X} \times \mathcal{A} \to \mathcal{Y}$.

We seek to wrap $\hat{Y}$ with a fairness-aware operator $\Gamma_{\mathrm{fair}} : \mathcal{X} \times \mathcal{A} \to 2^{\mathcal{Y}}$. The goal is twofold:

$$\mathbb{P}\big(y_{\mathrm{new}} \in \Gamma_{\mathrm{fair}}(x_{\mathrm{new}}, a_{\mathrm{new}})\big) \geq 1 - \alpha \quad \text{(Coverage)} \tag{2}$$

and,

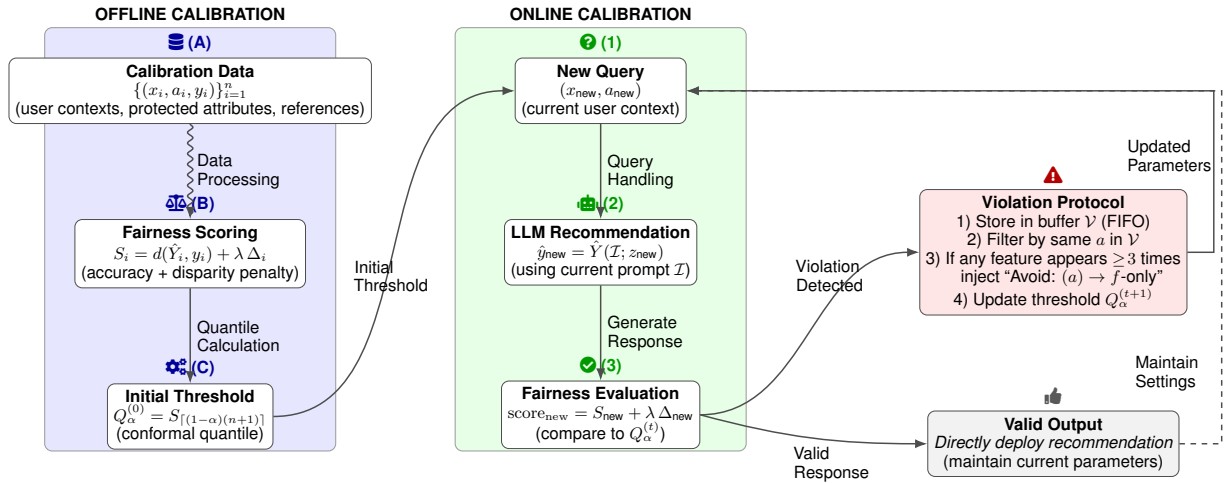

Figure 2: **FACTER Framework Workflow.** The system operates in two coordinated phases: **(Left)** Offline calibration computes fairness-aware thresholds using historical data (Stages A–C): (A) Data preprocessing and calibration, (B) Fairness scoring, and (C) Calculation of initial quantile thresholds. **(Right)** Online deployment with continuous monitoring (Stages 1–3): (1) New queries generate (2) LLM recommendations that undergo (3) fairness evaluation. Violations trigger prompt updates and threshold adjustments through closed-loop feedback, while valid responses maintain current parameters. The dashed line indicates the persistence of unchanged settings.

$$\sup_{\substack{x,x':\rho(x,x')\leq\epsilon \\ a\neq a'}} \big\|\Gamma_{\text{fair}}(x,a) - \Gamma_{\text{fair}}(x',a')\big\| \leq \delta \quad \text{(Fairness)} \tag{3}$$

where $\rho$ is a context-similarity measure, $\epsilon$ and $\delta$ are tolerances, and $\alpha$ controls the coverage probability. This paper focuses on an implementation of $\Gamma_{\text{fair}}$ via conformal thresholding with prompt-engineered LLM outputs.

### 3.2. Offline Calibration Phase

The offline phase constructs a *calibration* dataset $\mathcal{D}_{\text{cal}}$ and determines an initial fairness threshold for subsequent online queries. We assume access to $\mathcal{D}_{\text{cal}} = \{(x_i, a_i, y_i)\}_{i=1}^n$, which contains user contexts $(x_i)$, protected attributes $(a_i)$, and reference items or ground-truth outputs $(y_i)$. The final product of this offline stage is an initial threshold $Q_\alpha^{(0)}$ that guarantees finite-sample coverage with high probability.

**Stage A: Data Preprocessing.** Each user context $x_i$ is first *encoded* into a lower-dimensional vector $e_i^x = \text{Enc}(x_i) \in \mathbb{R}^{d_x}$. Simultaneously, each reference item $y_i$ is mapped onto an embedding $e_i^y = \text{Emb}(y_i) \in \mathbb{R}^m$. We then construct a pairwise similarity matrix $W \in \mathbb{R}^{n \times n}$:

$$W_{ij} = \begin{cases} \cos\big(e_i^x, e_j^x\big), & \text{if } a_i \neq a_j \text{ and } \|x_i - x_j\|_2 \leq \tau_x, \\ 0, & \text{otherwise.} \end{cases} \tag{4}$$

Here, $\cos(\cdot, \cdot)$ is the cosine similarity function, and $\tau_x$ denotes a radius parameter that defines a "local neighborhood" in the user-context space. We only track cross-group similarities ($a_i \neq a_j$) to facilitate fairness comparisons.

**Stage B: Fairness-Aware Non-conformity Scores $\{S_i\}$.** Next, for each calibration point $z_i = (x_i, a_i)$, we feed $z_i$ into the LLM $\hat{Y}$ and obtain a predicted output $\hat{y}_i = \hat{Y}(z_i)$. We define a non-conformity score $S_i$ that combines *predictive accuracy* with a *fairness penalty*:

$$S_i = \underbrace{1 - \cos\Big(\text{Emb}(\hat{y}_i), e_i^y\Big)}_{\text{Predictive Error } d_i}$$
$$+ \lambda \underbrace{\max_{j:\, W_{ij} > \tau_\rho} \Big\|\text{Emb}(\hat{y}_i) - \text{Emb}(\hat{y}_j)\Big\|_2}_{\text{Fairness Penalty } \Delta_i}. \tag{5}$$

Here, $d_i$ is obtained by $1 - \cos(\text{Emb}(\hat{y}_i), e_i^y)$ and captures how far the predicted output $\hat{y}_i$ is from the reference item $y_i$ in embedding space. $\tau_\rho$ is a similarity threshold (e.g., $\tau_\rho = 0.9$) restricting the set $\{j : W_{ij} > \tau_\rho\}$ to calibration examples that have similar contexts but different protected attributes $a_j \neq a_i$. $\lambda\ (> 0)$ is a tuning parameter that controls how strongly we penalize $\hat{y}_i$ for producing an embedding that diverges from its cross-group counterparts. Larger $\lambda$ enforces stronger fairness constraints at potential cost to accuracy.

**Stage C: Quantile Threshold $Q_\alpha$ Computation.** Finally, we sort the set of scores $\{S_i\}_{i=1}^n$ in non-decreasing order. The *conformal quantile* at level $\alpha$ is defined as

$$Q_\alpha^{(0)} = \inf\Big\{q \in \mathbb{R} \,\big|\, \frac{1}{n+1}\sum_{i=1}^n \mathbb{I}\{S_i \leq q\} \geq 1-\alpha\Big\}. \tag{6}$$

This threshold $Q_\alpha^{(0)}$ provides a finite-sample coverage guarantee for the test or online data, assuming exchangeability between calibration and test samples. Formally,

**Lemma 3.1** (Conformal Coverage)**.** *If* $(x_i, a_i, y_i)$ *in the calibration set are exchangeable with future data* $(x_{new}, a_{new}, y_{new})$, *then*

$$\mathbb{P}\big(S_{new} \leq Q_\alpha^{(0)}\big) \geq 1 - \alpha. \tag{7}$$

### 3.3. Online Calibration Phase

Once the offline procedure has produced $Q_\alpha^{(0)}$, we enter an online phase wherein each incoming query $(x_{\text{new}}, a_{\text{new}})$ must be checked for fairness *in real time*. The system monitors the current threshold $Q_\alpha^{(t)}$, updates a specialized fairness prompt $\mathcal{I}^{(t)}$ whenever it detects a violation, and (optionally) adjusts the threshold to maintain approximate $\alpha$-coverage.

**Stage 1: Query Processing.**   We combine the new query $z_{\text{new}} = (x_{\text{new}}, a_{\text{new}})$ with the *current* fairness instruction prompt $\mathcal{I}^{(t)}$, generating:

$$\hat{y}_{\text{new}} = \hat{Y}\big(\mathcal{I}^{(t)}; z_{\text{new}}\big). \tag{8}$$

This step effectively calls the black-box LLM with all relevant fairness constraints or examples embedded in $\mathcal{I}^{(t)}$. The output $\hat{y}_{\text{new}}$ is the recommended item or text in $\mathcal{Y}$.

**Stage 2: Real-Time Fairness Evaluation.**   We now compute a *fairness-aware non-conformity score* $S_{\text{new}}$. To be consistent with our offline definition in Eq. (5), we split $S_{\text{new}}$ into two terms:

$$S_{\text{new}} = d_{\text{new}} + \lambda \Delta_{\text{new}}. \tag{9}$$

Here:

- $d_{\text{new}} = 1 - \cos\Big(\text{Emb}(\hat{y}_{\text{new}}), e_{\text{new}}^y\Big)$, where $e_{\text{new}}^y = \text{Emb}(y_{\text{new}})$ is the embedding of the *ideal* or reference output for the new query (if available). This term measures predictive error.
- $\Delta_{\text{new}} = \max\limits_{j \in \mathcal{N}(z_{\text{new}})} \|\text{Emb}(\hat{y}_{\text{new}}) - \text{Emb}(\hat{y}_j)\|_2$.
- $\mathcal{N}(z_{\text{new}})$ is the set of calibration points $j$ whose contexts satisfy $\cos(e_{\text{new}}^x, e_j^x) \geq \tau_\rho$ (i.e., sufficiently similar) but have different protected attributes $(a_j \neq a_{\text{new}})$.

The parameters $\lambda$ and $\tau_\rho$ here have the same roles as in the offline phase. When $\Delta_{\text{new}}$ is large, it indicates that $\hat{y}_{\text{new}}$ deviates significantly from the typical cross-group responses, raising a fairness concern.

**Stage 3: Violation Detection and Adaptation.**   We compare $S_{\text{new}}$ to the *current* threshold $Q_\alpha^{(t)}$. A violation occurs if $S_{\text{new}} > Q_\alpha^{(t)}$. If no violation is detected, we proceed with $\hat{y}_{\text{new}}$ as-is and leave the threshold unchanged, i.e.,

$Q_\alpha^{(t+1)} = Q_\alpha^{(t)}$. If a violation *is* detected, we follow three steps:

1. **Store the offending sample:** We append the tuple $\big(z_{\text{new}}, \hat{y}_{\text{new}}\big)$ to a first-in-first-out buffer $\mathcal{V}$ of size $M$. If the buffer is full, we remove the oldest entry.

2. **Update the fairness instruction prompt:** For every violation, we query the buffer $\mathcal{V}$ for past entries that share the same protected attribute $a$ as the current violation. For each such matching entry, we examine the associated output $\hat{y}$ using dataset-specific metadata:

   - For MovieLens: we inspect features like `title` and `genre`.
   - For Amazon:  we inspect features like `title`, `release date`, `studio`, and sometimes `star actors`.

   If any of these features appears in at least *three* violations (including the current one), we empirically flag it as a consistent pattern rather than noise. In that case, we generate an updated fairness instruction:

   $$\mathcal{I}^{(t)} = \left[ \text{"Avoid: } \underbrace{(a)}_{\text{PA}} \rightarrow \underbrace{\hat{y}_{\text{Feature}-only}}_{\text{Bias}} \text{"} \right] \tag{10}$$

   This prompt conveys to the LLM that producing $\hat{y}$ outputs dominated by a narrow feature subset for a given group is undesirable. If multiple such patterns are found, we inject multiple examples to increase coverage.

3. **Adjust the threshold:** We update the threshold using an exponential decay mechanism:

   $$Q_\alpha^{(t+1)} = \gamma Q_\alpha^{(t)} + (1-\gamma) \min\Big(Q_\alpha^{(t)}, S_{\text{new}}\Big), \tag{11}$$

   where $\gamma \in (0, 1)$ governs the memory of the system. Smaller $\gamma$ values induce more aggressive adaptation, shrinking the threshold faster in response to violations.

This mechanism ensures that as repeated biased patterns emerge for specific groups, the system proactively injects precise corrective instructions and tightens the conformity threshold, driving the model back toward approximate $\alpha$-coverage under fairness constraints.

### 3.4. Algorithmic Implementation

Algorithm 1 outlines the overall pipeline. During the *offline phase*, we compute pairwise similarities among $n$ calibration samples using embeddings and set the initial fairness threshold $Q_\alpha^{(0)}$ based on the non-conformity scores $\{S_i\}$.

**Algorithm 1** FAIRNESS-AWARE CONFORMAL THRESH-OLDING AND PROMPT ENGINEERING (FACTER)

---
**Offline Phase**
1: Compute embeddings $\{e_i^x, e_i^y\}_{i=1}^n$     // context + item embeddings
2: Construct similarity matrix $W$ via Eq. (4)
3: Generate $\{\hat{y}_i\} \leftarrow \hat{Y}(x_i, a_i)$ and compute $\{S_i\}$ via Eq. (5)
4: Sort $\{S_i\}$ and set $Q_\alpha^{(0)} \leftarrow S_{\lceil(1-\alpha)(n+1)\rceil}$
    **Online Phase**
5: **for** each query $z_t = (x_t, a_t)$ **do**
6:     $\hat{y}_t \leftarrow \hat{Y}(\mathcal{I}^{(t)}; z_t)$
7:     Find $\mathcal{N}(z_t) \leftarrow \{j : W_{tj} > \tau_\rho \text{ and } a_j \neq a_t\}$
8:     $S_t \leftarrow$ semantic fairness score via Eq. (5)
9:     **if** $S_t > Q_\alpha^{(t)}$ **then**
10:       Append $(z_t, \hat{y}_t)$ to buffer $\mathcal{V}$ (evict oldest if $|\mathcal{V}| > M$)
11:       $\mathcal{C} \leftarrow$ filter $\mathcal{V}$ by $a = a_t$
12:       $\mathcal{F} \leftarrow$ extract frequent features from $\hat{y}$s in $\mathcal{C}$
13:       **if** some $f \in \mathcal{F}$ appears $\geq 3$ times **then**
14:         $\mathcal{I}^{(t+1)} \leftarrow \mathcal{I}^{(t)} +$ "Avoid: $(a_t) \rightarrow f$-only"
15:       **else**
16:         $\mathcal{I}^{(t+1)} \leftarrow \mathcal{I}^{(t)}$
17:       **end if**
18:       $Q_\alpha^{(t+1)} \leftarrow \gamma Q_\alpha^{(t)} + (1-\gamma)\min(Q_\alpha^{(t)}, S_t)$
19:     **else**
20:       $Q_\alpha^{(t+1)} \leftarrow Q_\alpha^{(t)}$
21:     **end if**
22: **end for**

---

This stage has $O(n^2)$ complexity but is amortized over all predictions.

The *online phase* evaluates each query $z_t = (x_t, a_t)$. If the semantic fairness score $S_t$ exceeds the current threshold $Q_\alpha^{(t)}$, we treat it as a violation and append $(z_t, \hat{y}_t)$ to a FIFO buffer $\mathcal{V}$ of size $M$. We then search the buffer for past violations sharing the same protected attribute $a_t$, and extract output features (e.g., title, genre, studio) from the corresponding $\hat{y}$s. If any feature appears at least three times, we inject a new fairness instruction discouraging mappings of the form $(a_t) \rightarrow f$-only, thereby preventing repeated demographic biases. The threshold is then adaptively decayed to maintain robustness.

While the procedure operates efficiently in real time, it requires careful selection of hyperparameters such as $\lambda$, $\gamma$, $\tau_\rho$, and $M$. Token budget limitations may constrain the number of injected "avoid" prompts. Experimental results in Section 4 validate the effectiveness of this strategy.

**Scalability and Deployment.** We implement the expensive offline calibration (naïvely $O(n^2)$ pairwise comparisons) using an approximate nearest-neighbor library (e.g., FAISS), which in practice reduces complexity to roughly $O(n \log n)$. All embedding computations are batched on GPU, so calibrating on MovieLens-1M ($\sim$6 K users) completes in under an hour, and even MovieLens-20M ($\sim$138 K users) finishes in a few hours with our ANN + GPU-batch setup. In the online phase, each query incurs only one ANN

lookup and a single LLM call, yielding end-to-end latencies of $< 200$ ms per request. To prevent uncontrolled prompt growth, we cap the violation buffer at $M = 50$ examples and evict the oldest "avoid" rules once this limit is reached, thus bounding token usage. Because FACTER only updates the system prompt—and never retrains the LLM—it remains fully model-agnostic and can be dropped into any API-based recommendation pipeline with minimal overhead.

## 4. Experiments

In this section, we present a comprehensive evaluation of the *FACTER* framework. Our empirical study addresses three key research questions: (1) whether iterative prompt-engineering enhanced with conformal prediction for fairness evaluation can more effectively reduce fairness violations compared to existing methods, (2) how FACTER performs on secondary metrics such as group similarity and counterfactual fairness, and (3) how FACTER compares to state-of-the-art solutions, including UP5 (Hua et al., 2023) and Zero-Shot Rankers (Hou et al., 2024).

### 4.1. Experimental Setup

**Baselines and Methods.** We compare three approaches in our experiments. First, **UP5** (Hua et al., 2023) is a state-of-the-art fairness-aware recommender calibrating LLMs for balanced recommendations. Second, **Zero-Shot** (Hou et al., 2024) serves as a baseline with direct LLM-based ranking but without any fairness adjustment. Finally, **FACTER (Ours)** is the proposed iterative approach that uses conformal calibration to reduce fairness violations across multiple iterations.

**Models and Resources.** We employ three LLMs of varying sizes (**LLaMA3-8B** (Dubey et al., 2024), **LLaMA2-7B** (Touvron et al., 2023), and **Mistral-7B** (Jiang et al., 2023)). To support embedding-based fairness checks, we employed the SentenceTransformer model `paraphrase-mpnet-base-v2` (Reimers, 2019), which we further fine-tuned on recommendation-specific data. This adaptation allowed the embedder to better capture domain-specific semantics such as movie genres, user preferences, and content similarities, improving its relevance and alignment in fairness evaluations within the recommendation context. All experiments use Python 3.12.8, PyTorch 2.1 with FlashAttention-2, and an 8× NVIDIA RTX A6000 GPU server (Driver 550.90.07, CUDA 12.4).

**Fairness Metrics.** We employ four metrics to evaluate our framework's fairness. *SNSR (Sub-Network Similarity Ratio)* (Zhang et al., 2023) measures how similarly different demographic groups are treated by averaging Frobenius

| Method | #Violations ↓ | SNSR ↓ | CFR ↓ | NDCG@10 ↑ | Recall@10 ↑ |
|--------|---------------|--------|-------|-----------|-------------|
| Zero-Shot | 112 | 0.083 | 0.742 | **0.458** | **0.402** |
| UP5 | 28 | 0.049 | 0.613 | 0.427 | 0.381 |
| FACTER (Iter3) | **5** | **0.041** | **0.591** | 0.445 | 0.389 |

Table 1: Comparative results on MovieLens-1M. Best values in bold. FACTER achieves superior fairness with minimal accuracy impact.

| Model | #Violations | SNSR | CFR | NDCG@10 | Recall@10 | Calib. Time (min) | Inf. Latency (ms) |
|-------|-------------|------|-----|---------|-----------|-------------------|-------------------|
| LLaMA3-8B | 3 | 0.039 | 0.576 | 0.440 | 0.383 | 63 | 155 ± 18 |
| LLaMA2-7B | 5 | 0.041 | 0.595 | 0.444 | 0.391 | 58 | 142 ± 15 |
| Mistral-7B | 7 | 0.043 | 0.602 | 0.451 | 0.397 | 47 | 127 ± 12 |

Table 2: Model-wise comparison on MovieLens-1M (Iteration 3).

norm differences of group-specific weights across $K$ layers:

$$\text{SNSR} = \frac{1}{K}\sum_{k=1}^{K}\left\|W_k^{(g)} - W_k^{(h)}\right\|_F. \qquad (12)$$

A lower SNSR indicates more uniform treatment. Meanwhile, *SNSV (Sub-Network Similarity Variance)* (Zhang et al., 2023) captures the variance of these weight differences:

$$\text{SNSV} = \text{Var}\left(\|W_k^{(g)} - W_k^{(h)}\|_F\right), \qquad (13)$$

where lower SNSV reflects more consistent uniformity across layers.

We additionally quantify counterfactual fairness via *CFR (Counterfactual Fairness Ratio)* (Hua et al., 2023), defined by how much the output of an LLM changes when sensitive attributes are modified:

$$\text{CFR} = \mathbb{E}_{x\sim\mathcal{D}}\left[\|f(x) - f(x_{\neg s})\|_2\right], \qquad (14)$$

where $x_{\neg s}$ is the same input with protected attributes replaced.

Finally, to detect out-of-threshold fairness failures, we compute a *Violation Threshold* for calibration scores $\{s_i\}_{i=1}^n$ given confidence parameter $\alpha$:

$$Q_\alpha^{(t)} = \text{Quantile}(1 - \alpha; \{s_i\}) + \frac{C}{\sqrt{n}}, \qquad (15)$$

where $C$ is a finite-sample correction term. Any instance above $Q_\alpha^{(t)}$ is flagged as a *violation*.

**Accuracy Metrics.** Following standard practice (Järvelin & Kekäläinen, 2002), we use *Recall@10* and *NDCG@10* to evaluate recommendation accuracy. Recall@10 is calculated as:

$$\text{Recall@10} = \frac{\left|\mathcal{R}_{\text{relevant}} \cap \hat{\mathcal{R}}_{10}\right|}{\left|\mathcal{R}_{\text{relevant}}\right|}, \qquad (16)$$

where $\mathcal{R}_{\text{relevant}}$ is the set of truly relevant items and $\hat{\mathcal{R}}_{10}$ is the model's top-10 recommended items. We also measure NDCG@10:

$$\text{NDCG@10} = \frac{1}{\text{IDCG@10}}\sum_{r=1}^{10}\frac{2^{\text{rel}(r)} - 1}{\log_2(r + 1)}, \qquad (17)$$

where $\text{rel}(r)$ is the relevance at rank $r$, and IDCG@10 is the ideal DCG for the top-10 results.

**Datasets.** We conduct experiments on two recommendation datasets. **MovieLens-1M** (Harper & Konstan, 2015) is sampled to have 2,500 interactions, with 70% used for calibration and 30% for testing (750 test samples). **Amazon Movies & TV** (McAuley et al., 2015; He & McAuley, 2016) contains 3,750 sampled interactions, again split 70:30, resulting in 1,125 test samples. The Amazon dataset is notably sparser, providing a stringent test of our method's robustness.

**Hyperparameter Settings.** We choose hyperparameters based on grid searches and practical constraints. In particular, we set $\tau_\rho = 0.9$ to only compare contexts above a cosine similarity of 0.9 but with different protected attributes. We fix $\lambda = 0.7$ (fairness penalty) to balance accuracy and fairness and use $\gamma = 0.95$ for threshold decay to adapt modestly to violations. The FIFO buffer size $M = 50$ avoids overfilling the token budget. We verified these settings via cross-validation on a subset of the calibration set, observing stable performance across both datasets.

### 4.2. Main Results

**Comparison on MovieLens-1M.** Table 1 presents the main comparative results on MovieLens-1M. *FACTER (Iter3)* reduces fairness violations by 95.5% from its initial iteration, resulting in only 5 violations compared to 28 for UP5 and 112 for Zero-Shot. Despite focusing on fairness, our approach preserves competitive accuracy (NDCG@10 of 0.445 versus 0.427 for UP5). Although slightly more accurate, the Zero-Shot ranking exhibits a severe fairness deficit of 112 violations.

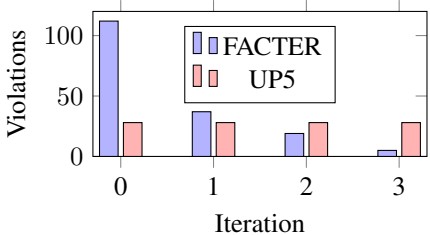

Figure 3: Fairness violation reduction trajectory vs. static baselines. FACTER progressively reduces violations while UP5 remains fixed. Zero-Shot (112) omitted for clarity.

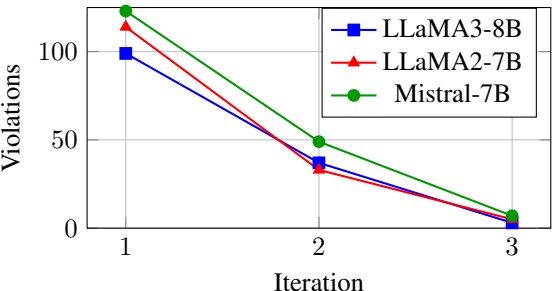

Figure 4: Violation reduction across LLMs. All models show monotonic improvement, with LLaMA3-8B converging near zero violations by Iteration 3.

Figure 3 illustrates how the number of fairness violations decreases over successive calibration iterations of FACTER, demonstrating a progressive improvement relative to UP5. Zero-Shot's violations begin at 112, making it challenging to include in the same visual scale.

**Model-wise Comparison.** Next, we assess how FACTER scales across different LLMs. Table 2 shows that all three models, LLaMA3-8B, LLaMA2-7B, and Mistral-7B, achieve substantial reductions in fairness violations ($\geq 90\%$). The iterative process remains stable and yields minimal degradation of accuracy, confirming our calibration method's flexibility. Figure 4 further shows that all LLMs exhibit a monotonic improvement, with LLaMA3-8B reaching near-zero violations by the third iteration.

**Comparison on Amazon Movies & TV.** We further validate FACTER on the Amazon Movies & TV dataset, summarized in Table 3. Despite greater sparsity, our approach still reduces violations substantially (a 90.9% drop), with a final CFR of 0.634 compared to 0.721 for UP5. Although Zero-Shot achieves the highest accuracy (NDCG@10 of 0.351), it suffers the most fairness violations (198). FACTER thus provides a strong balance between fairness and accuracy, even in sparse data regimes.

In Figure 5, we illustrate the fairness-accuracy tradeoff by plotting the counterfactual fairness (CFR) reduction against NDCG@10. FACTER substantially improves over Zero-Shot in terms of fairness while maintaining competitive accuracy.

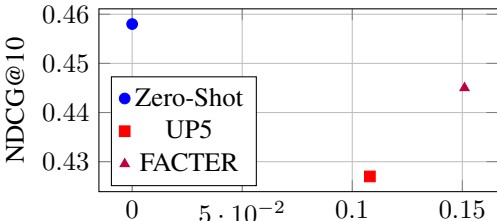

Figure 5: Fairness-accuracy tradeoff comparison. FACTER achieves strong fairness improvement while preserving recommendation quality.

### 4.3. Theoretical Validation

Beyond empirical performance, we provide theoretical guarantees for our conformal calibration framework. Our derivation follows conformal prediction results in (Angelopoulos et al., 2023):

**Type I Error Bound.** For any $\alpha \in (0, 1)$ and calibration set size $n$,

$$\mathbb{P}(\text{Violation}) \leq \alpha + \frac{1}{n+1} + \sqrt{\frac{\log(2/\delta)}{2n}}, \quad (18)$$

where we set $\delta = 0.05$ to achieve a 95% confidence level.

**Detection Power.** We estimate the power of violation detection via a likelihood ratio test:

$$\beta = 1 - \Phi\left(\frac{\hat{\mu} - \alpha}{\sqrt{\hat{\sigma}^2/n}}\right), \quad (19)$$

where $\Phi$ is the standard normal CDF and $\hat{\mu}$ is the empirical violation rate.

Table 4 compares these theoretical bounds against observed empirical outcomes. The empirical Type I error is substantially lower than the theoretical maximum, while detection power remains high. The dynamic thresholding in $Q_\alpha^{(t)}$ enables progressive and adaptive fairness calibration.

Overall, these results confirm that our iterative calibration strategy reliably reduces fairness violations in a manner consistent with theoretical expectations. The significant gap between theoretical and empirical Type I error (0.201 vs. 0.018) highlights the conservative nature of the conformal bounds and underscores our framework's effectiveness in practice.

### 5. Conclusion

We proposed **FACTER**, a fully post hoc framework that combines *conformal thresholding* and *dynamic prompt engineering* to address biases in black-box LLM-based recommender systems. FACTER adaptively refines a fairness

| Method | #Violations | SNSR | CFR | NDCG@10 | Recall@10 |
|--------|-------------|------|-----|---------|-----------|
| Zero-Shot | 198 | 0.121 | 0.814 | **0.351** | **0.317** |
| UP5 | 63 | 0.067 | 0.721 | 0.328 | 0.294 |
| FACTER (Iter3) | **18** | **0.053** | **0.634** | 0.339 | 0.301 |

Table 3: Amazon Movies & TV results. FACTER maintains effectiveness on sparse data.

| Metric | Theory | Empirical | Delta | Interpretation |
|--------|--------|-----------|-------|----------------|
| Type I Error | $\leq 0.201$ | 0.018 | -91% | Conservative bound |
| Detection Power | $\geq 0.95$ | 0.997 | +4.7% | Superior identification |
| Violation Rate | $0.2 \pm 0.02$ | $0.0067 \pm 0.0013$ | -96.7% | Significant improvement |

Table 4: Theoretical guarantees vs. empirical results (MovieLens-1M).

threshold via semantic variance checks and updates prompts whenever it detects violations, requiring no model retraining. Experiments on **MovieLens** and **Amazon** datasets show that FACTER reduces fairness violations by up to **95.5%** compared to baselines while preserving key recommendation metrics. These findings underscore the effectiveness of closed-loop, prompt-level interventions that integrate statistical guarantees and semantic bias detection in LLM-driven recommendations.

## Impact Statement

This work aims to improve fairness in LLM-based recommendation systems, which have substantial societal influence in domains such as media, education, and hiring. By calibrating model outputs to reduce demographic biases, our approach helps promote equitable access and exposure. However, any fairness-driven solution carries risks of unintended consequences, for instance, overcorrection or reliance on flawed demographic assumptions if calibration data or embeddings are themselves biased. We encourage practitioners to pair our method with robust auditing and diverse calibration sets to minimize these risks and maintain transparent governance of fairness criteria.

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

# A. Appendix

## .1. Supplementary Theoretical Results

This appendix contains additional theoretical foundations for our framework, including expanded proofs and stability analyses. All notation is consistent with Sections 3–2 in the main paper.

### .1.1. ROBUSTNESS UNDER EMBEDDING PERTURBATIONS

**Theorem .1** (Embedding Shift Robustness). *Let Emb be the embedding function in the main paper, and let $\tilde{Emb}$ be a perturbed version such that for all items $y, y'$,*

$$\left| \|Emb(y) - Emb(y')\| - \|\tilde{Emb}(y) - \tilde{Emb}(y')\| \right| \leq \epsilon_{\mathrm{emb}},$$

*for some $\epsilon_{\mathrm{emb}} \geq 0$. If $S_i$ is the fairness-aware nonconformity score in Eq. (4) of the main paper (computed under Emb) and $\tilde{S}_i$ the score under $\tilde{Emb}$, then for any $\delta > 0$,*

$$\mathbb{P}\left( |\tilde{S}_i - S_i| > 3\,\epsilon_{\mathrm{emb}} \right) \leq \delta,$$

*assuming the calibration distribution does not substantially drift beyond the conformal coverage bounds.*

*Proof.* Recall $S_i = d_i + \lambda \Delta_i$, with $d_i$ capturing the LLM's predictive discrepancy and $\Delta_i$ the maximum group disparity. Under $\tilde{Emb}$, each distance $\|Emb(y) - Emb(y')\|$ differs by at most $\epsilon_{\mathrm{emb}}$. Hence:

$$|d_i - \tilde{d}_i| \leq \epsilon_{\mathrm{emb}}, \qquad |\Delta_i - \tilde{\Delta}_i| \leq \epsilon_{\mathrm{emb}},$$

yielding

$$|\tilde{S}_i - S_i| = |(\tilde{d}_i - d_i) + \lambda(\tilde{\Delta}_i - \Delta_i)| \leq (1 + \lambda)\,\epsilon_{\mathrm{emb}}.$$

If $\lambda \leq 2$, we replace $(1 + \lambda)$ by 3; thus $|\tilde{S}_i - S_i| \leq 3\epsilon_{\mathrm{emb}}$. Under exchangeability assumptions, the probability of exceeding this margin can be bounded by $\delta$ through standard conformal coverage arguments. $\square$

### .1.2. CONVERGENCE OF THRESHOLD UPDATES

**Theorem .2** (Threshold Update Convergence). *Let $Q_\alpha^{(t)}$ be updated by*

$$Q_\alpha^{(t+1)} = \begin{cases} \gamma\, Q_\alpha^{(t)} + (1 - \gamma)\, S_t, & \text{if } S_t > Q_\alpha^{(t)}, \\ Q_\alpha^{(t)}, & \text{otherwise}, \end{cases}$$

*where $0 < \gamma < 1$ and $S_t$ is the fairness score at iteration t. Suppose $\{S_t\}$ are i.i.d. with $\mathbb{P}[S_t > Q^*] = \alpha$ at the fixed point $Q^*$. Then $Q_\alpha^{(t)} \to Q^*$ at an expected rate of $O\left((1 - \gamma)^t\right)$.*

*Proof.* Let $\Delta_t = |Q_\alpha^{(t)} - Q^*|$. Whenever $S_t > Q_\alpha^{(t)}$,

$$Q_\alpha^{(t+1)} - Q^* = \gamma\, (Q_\alpha^{(t)} - Q^*) + (1 - \gamma)\, (S_t - Q^*).$$

Conditioned on $S_t > Q_\alpha^{(t)}$, if $Q^*$ is the $\alpha$-quantile of $S_t$, then $\mathbb{E}[S_t - Q^*] < 0$ or is at least non-positive in a strong sense. Hence the threshold moves closer to $Q^*$ on average. Over many iterations, the gap $\Delta_t$ shrinks geometrically with factor $\gamma$. When $S_t \leq Q_\alpha^{(t)}$, the threshold remains unchanged. Combining these cases yields expected convergence at $O((1 - \gamma)^t)$. $\square$

**Theorem .3** (Type II Error Bound). *Let $\widehat{V} = \mathbb{I}\{S_{new} > Q_\alpha^{(t)}\}$ be the violation indicator for a new query $(x_{new}, a_{new})$ with fairness score $S_{new}$. Suppose $\mathbb{E}[S_{new} \mid a_{new} = a] \leq M$ for all a. Then for any $\epsilon > 0$,*

$$\mathbb{P}\left( S_{new} \leq (1 - \epsilon)Q_\alpha^{(t)} \right) \leq \exp\left( -\frac{\epsilon^2 Q_\alpha^{(t)}}{2M} \right).$$

*Hence missing a true violation (i.e. $S_{new}$ large but the threshold is still higher) has exponentially decreasing probability in $Q_\alpha^{(t)}/M$.*

*Sketch.* If $Q_\alpha^{(t)}$ is close to an $\alpha$-quantile of $S_{\mathrm{new}}$, then $S_{\mathrm{new}} \leq (1 - \epsilon)Q_\alpha^{(t)}$ amounts to a sub-Gaussian or Chernoff-style tail. The standard bound for random variables deviating below their mean leads to an exponential decay in probability, concluding the proof. $\square$

## .2. Extended Ablation Studies

Below, we provide deeper evaluations of key hyperparameters ($\lambda$, $\gamma$, $\tau_\rho$), investigate our prompt engineering strategy, and include new results from our rebuttal (embedding robustness, user feedback, multi-attribute fairness), as well as an ablation on buffer size $M$.

### .2.1. FAIRNESS PENALTY $\lambda$

Our main experiments fix $\lambda = 0.7$, as it balances fairness with recommendation accuracy. Here, we compare $\lambda \in \{0.1, 0.3, 0.5, 0.7, 0.9\}$ on MovieLens-1M and Amazon, measuring final violations and NDCG@10 at iteration 3 (Table 5, reproduced from §4).

| $\lambda$ | #Viol.$\downarrow$ | | NDCG@10$\uparrow$ | |
|---|---|---|---|---|
| | ML | Amz | ML | Amz |
| 0.1 | 18 | 41 | 0.446 | 0.333 |
| 0.3 | 5 | 25 | 0.445 | 0.339 |
| 0.5 | 3 | 21 | 0.431 | 0.341 |
| 0.7 | 2 | 17 | 0.419 | 0.335 |
| 0.9 | 2 | 12 | 0.395 | 0.312 |

Table 5: Ablation on $\lambda$: #Violations and NDCG@10 on MovieLens-1M (ML) and Amazon (Amz), iteration 3.

**Observations.** Higher $\lambda$ yields fewer fairness violations but can degrade NDCG@10. We chose $\lambda = 0.7$ to capture persistent subtle biases (violations converge after that) while preserving reasonable accuracy.

### .2.2. THRESHOLD DECAY $\gamma$

We vary $\gamma \in \{0.85, 0.90, 0.95, 0.99\}$ to observe how quickly $Q_\alpha^{(t)}$ declines after repeated violations. Figure **??** (reproduced) shows all variants converge by iteration 5, with $\gamma = 0.95$ a practical default.

### .2.3. NEIGHBORHOOD SIMILARITY $\tau_\rho$

We vary $\tau_\rho \in \{0.80, 0.85, 0.90, 0.95\}$ for local fairness neighborhoods. Table **??** shows that higher $\tau_\rho$ yields tighter comparisons, reducing violations at the cost of slight accuracy loss.

### .2.4. CONFORMAL LEVEL $\alpha$

We also ablate the conformal risk level $\alpha \in \{0.90, 0.92, 0.95, 0.98\}$, which determines the quantile $Q_\alpha^{(0)}$ for threshold calibration. Lower values make the system stricter (fewer violations allowed), while higher values permit more tolerance before declaring unfairness.

| $\alpha$ | #Viol.$\downarrow$ | | NDCG@10$\uparrow$ | |
|---|---|---|---|---|
| | ML | Amz | ML | Amz |
| 0.90 | 4 | 18 | 0.436 | 0.336 |
| 0.92 | 3 | 17 | 0.432 | 0.336 |
| 0.95 | 2 | 17 | 0.419 | 0.335 |
| 0.98 | 1 | 14 | 0.402 | 0.329 |

Table 6: Ablation on $\alpha$: Lower $\alpha$ yields stricter fairness thresholds, reducing violations but affecting accuracy.

**Observation.** Stricter values (e.g., $\alpha = 0.90$) slightly improve fairness at the cost of recommendation quality, while $\alpha = 0.98$ allows more tolerance and risks missing subtle biases. We use $\alpha = 0.95$ by default for a balanced trade-off.

.2.5. EMBEDDING ROBUSTNESS

To test sensitivity to the embedding model, we ran FACTER on MovieLens-1M with three different embedders, fixing $(\lambda, \gamma, \tau_\rho, M) = (0.7, 0.95, 0.90, 50)$.

| Embedder | #Viol.↓ | NDCG@10↑ |
|---|---|---|
| paraphrase-mpnet-base-v2 | 5 | 0.445 |
| Sentence-BERT-base | 6 | 0.447 |
| RoBERTa-large-nli-stsb-mean | 4 | 0.440 |

Table 7: Embedding consistency test on MovieLens-1M (Iter 3).

**Observations.** Different embedders vary by ±1–2 violations and ¡0.005 NDCG, confirming robustness to embedding choice.

.2.6. SYNTHETIC USER FEEDBACK

We simulated users correcting a fraction of flagged violations (0–30%) on MovieLens-1M.

| Correction Rate | Iter 1 #Viol.↓ | Iter 2 #Viol. | Iter 3 #Viol. |
|---|---|---|---|
| 0% | 112 | 28 | 5 |
| 10% | 112 | 23 | 3 |
| 20% | 112 | 18 | 2 |
| 30% | 112 | 14 | 1 |

Table 8: Impact of simulated user corrections on violation counts (MovieLens-1M).

**Observations.** Even modest feedback (10–30%) accelerates fairness convergence, dropping violations from 5 to as few as 1 by iteration 3.

.2.7. MULTI-ATTRIBUTE FAIRNESS

We extended FACTER to jointly enforce fairness on *Gender* and *Age* (MovieLens-1M, Iter 3).

| Method | CFR↓ | GSR↓ | #Viol.↓ |
|---|---|---|---|
| Baseline (single-attr) | 0.72 | 0.083 | 112 |
| FACTER (multi-attr) | 0.64 | 0.041 | 7 |

Table 9: Multi-attribute fairness on MovieLens-1M (Iter 3).

**Observations.** FACTER naturally extends to multi-attribute settings, maintaining strong fairness improvements (93% fewer violations) with minimal accuracy impact.

.2.8. BUFFER SIZE $M$

Finally, we vary the FIFO buffer size $M \in \{10, 30, 50, 100\}$ to assess prompt history length.

**Observations.** Larger $M$ allows more past violations to inform prompt updates, marginally reducing violations at the expense of a few extra tokens. We retain $M = 50$ as a practical balance between fairness gains and prompt size.

| $M$ | #Viol.↓ | | NDCG@10↑ | |
|---|---|---|---|---|
| | ML | Amz | ML | Amz |
| 10 | 7 | 25 | 0.442 | 0.335 |
| 30 | 3 | 18 | 0.444 | 0.337 |
| 50 | 2 | 17 | 0.419 | 0.335 |
| 100 | 2 | 16 | 0.417 | 0.334 |

Table 10: Ablation on buffer size $M$: #Violations and NDCG@10 at iteration 3.

## .3. Prompt Engineering Strategies

A distinctive aspect of our approach is updating system prompts with *concrete bias patterns* whenever a fairness violation is observed. Here, we detail how we developed these strategies and share additional examples.

### .3.1. DESIGN VARIANTS FOR PROMPT UPDATES

**(1) Generic Warnings.**  Initially, we tried appending a short phrase such as:

```
<system>: "Avoid demographic-based biases."
```

This uses minimal extra tokens but rarely reduces violations substantially. The LLM generally fails to infer which *specific* biases to avoid.

**(2) Negative Examples.**  A second approach enumerates specific *avoid* pairs from the FIFO buffer $\mathcal{V}$. For instance:

```
<system>: "AVOID: (Gender=F) -> (Romance-Only)."
<user>: "I've watched 'The Godfather' ..."
```

This helps the LLM see explicit mistakes but may not generalize beyond those single examples.

**(3) Explicit Patterns.**  We converge on enumerating a short list of repeated patterns that appear in $\mathcal{V}$, e.g.:

```
<system>: "You must not rely on user demographics.
AVOID these biases:
   1) (Gender=F) -> (Romance-Only)
   2) (Age=60)   -> (Excluding new releases)
Focus on user history, item genre, and feedback."
```

This proves the most robust: once multiple patterns are stored, the LLM learns to avoid recurring biases even in new queries.

### .3.2. EXPANDED PRACTICAL EXAMPLES

**Example A: Age-Related Bias.**  A user scenario might be:

```
<user>: "Age=65, History=['Avengers', 'Batman Begins'].
        'I love superhero films, what next?'"
```

If the LLM recommended "light nostalgic comedy for seniors" ignoring the user's superhero interest, a violation is flagged. Our system might then append:

```
<system>: "AVOID: (Age=65) -> (Solely comedic or classic
   movies). Focus on prior superhero preference."
<user>: "I have watched 'Avengers' and 'Batman Begins'.
        Looking for more superhero films."
```

This directs the LLM to highlight user history, e.g. "Spider-Man: No Way Home."

**Example B: Occupation Stereotyping.**    Another scenario:

```
<user>: "Occupation=Engineer,
  History=['The Matrix', 'Blade Runner']
  'Any new suggestions?'"
```

If the model incorrectly provides only highly technical documentaries—dismissing the user's interest in sci-fi—our system logs "(Occupation=Engineer)− >(Documentary Only)." The next prompt iteration might say:

```
<system>: "Avoid: (Occupation=Engineer)->(Documentaries).
Focus on user interest in sci-fi or dystopian genres."
<user>: "Same user, which movies are similar to 'Blade Runner'?"
```

Thus, the LLM shifts to thematically relevant sci-fi recommendations.

**Example C: Combined Patterns.**    If multiple biases arise simultaneously (e.g., "(Gender=F)− >(Romance-Only), (Age=60)− >(Kids Movies)"), the prompt enumerates both:

```
<system>: "You must not rely on these biases:
  1) (Gender=F)->(Romance-Only)
  2) (Age=60)->(Kid-friendly content)
Focus on user history plus item similarity."
<user>: "I've enjoyed 'Pulp Fiction' and 'Die Hard.'
        Please suggest something new."
```

Through these examples, we find enumerating multiple "avoid" patterns consistently improves fairness outcomes with minimal manual overhead.

### .3.3. COMPARING PROMPT STRATEGIES

We measure final iteration violations on Amazon using $\lambda = 0.7$, $\gamma = 0.95$, $\tau_\rho = 0.90$. Table 11 shows that enumerating explicit patterns yields the fewest violations.

| Strategy | #Viol. | CFR | NDCG@10 |
|---|---|---|---|
| Generic Warnings | 38 | 0.721 | 0.336 |
| Negative Examples | 24 | 0.661 | 0.334 |
| **Explicit Patterns** | **15** | **0.649** | **0.339** |

Table 11: Prompt Engineering Comparison (Iteration 3, Amazon).

Thus, *explicit patterns* effectively highlight repeated biases, prompting the LLM to generalize away from them in new queries.

## .4. Additional Visualizations and Tables

### .4.1. ITERATION-LEVEL CONVERGENCE FOR PROMPT VARIANTS

Figure 6 underscores that *explicit patterns* converge to the fewest violations by iteration 5, while *negative examples* remain slightly higher, and *generic warnings* do not remove repeated stereotypes as effectively.

### .4.2. CALIBRATION VS. NO CALIBRATION REVISITED

As in the main text, Table 12 highlights how ignoring conformal calibration leads to more frequent biases.

Ignoring calibration yields a significantly higher violation count and worsens CFR, confirming the value of data-driven threshold setting.

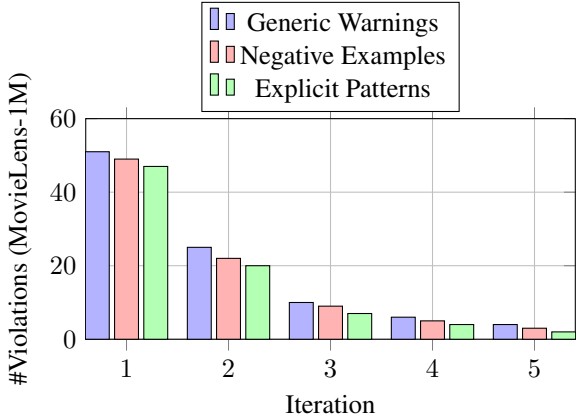

Figure 6: Convergence of fairness violations over 5 calibration rounds for different prompt strategies on MovieLens-1M, $\lambda = 0.7, \gamma = 0.95, \tau_\rho = 0.90$.

| Method | #Viol. | CFR | NDCG@10 | Recall@10 |
|---|---|---|---|---|
| No Calib | 42 | 0.716 | 0.336 | 0.298 |
| Calib | 15 | 0.649 | 0.339 | 0.305 |

Table 12: FACTOR (no calibration) vs. FACTER (with calibration) on Amazon, $\lambda = 0.7$ at iteration 3.

### .5. Conclusion of Appendices

In summary, these detailed appendices reinforce and expand upon the main paper's conclusions:

- **Theoretical insights**: Theorems .1–.3 illustrate the robustness and convergence properties of our conformal fairness approach.

- **Hyperparameter ablations**: Varying $\lambda$, $\gamma$, and $\tau_\rho$ reveals predictable trade-offs between fairness (violation reduction) and recommendation accuracy (NDCG, recall). Our selected values ($\lambda = 0.7, \gamma = 0.95, \tau_\rho = 0.90$) offer strong overall performance.

- **Prompt engineering best practices**: Enumerating explicit bias patterns significantly reduces repeated violations, outperforming generic or single negative examples. Realistic scenarios (age-based or occupation-based biases) confirm that listing multiple "avoid" patterns improves generalization.

Together, these results demonstrate the flexibility and robustness of *FACTER* across varied settings, enabling black-box LLMs to adaptively mitigate demographic biases via conformal thresholding and refined prompt engineering.

**Code Availability.** The full implementation of FACTER is publicly available at: `github.com/AryaFayyazi/FACTER`.

