# OpenReview forum: "FACTER: Fairness-Aware Conformal Thresholding and Prompt Engineering for Enabling Fair LLM-Based Recommender Systems"
_ICML.cc/2025/Conference — ICML 2025 poster_

### Official Review · Reviewer_D7oe · 2025-03-14

**Overall Recommendation:** 1

**Summary:**

The authors present a fairness-aware framework for LLM-based recommendation systems that combines conformal prediction with dynamic prompt engineering. FACTER introduces an adaptive semantic variance threshold and a violation-triggered mechanism to tighten fairness constraints when biases arise.

**Claims And Evidence:**

mostly supported

**Essential References Not Discussed:**

Some references are not included.

**Experimental Designs Or Analyses:**

Probably valid.

**Methods And Evaluation Criteria:**

probably sound

**Other Comments Or Suggestions:**

NA

**Other Strengths And Weaknesses:**

Strengths
1. Integrate Conformal Prediction for Fairness Calibration
Prior works on fairness in LLM-based recommendation [1] mainly rely on direct re-ranking or pretraining constraints. The paper leverages conformal prediction to define fairness violation thresholds, a statistically principled approach in LLM recommendations.

2. Effective Bias Mitigation Without Model Retraining
Unlike prior adversarial training-based methods (e.g., [2]), which require modifying model parameters, FACTER works in a black-box setting, making it suitable for API-based deployments (e.g., OpenAI, Hugging Face models).




Weaknesses
1. Incremental. The paper combines multiple existing techniques, such as conformal prediction and prompt engineering, but offers limited technical novelty.

2. Over-Reliance on Embedding-Based Fairness Measures
FACTER defines fairness violations using embedding distances (computed via Sentence-Transformers). This assumes that embeddings capture fairness-sensitive information correctly, which may not always be valid. Bias in embeddings themselves could affect fairness evaluations [3,4]. If the embeddings already encode demographic biases, fairness constraints based on them may be flawed.


3. Limited Justification for Fairness Threshold Selection
The paper claims to use conformal prediction for threshold calibration, but the choice of α (confidence level) lacks theoretical justification. The authors should conduct an ablation study comparing different threshold selection methods (e.g., data-driven quantile calibration vs. fixed conformal bounds).


4. Scalability Issues in Large-Scale Deployments
FACTER’s offline calibration phase has O(n²) complexity, requiring pairwise similarity comparisons across all calibration points. This makes it computationally expensive for large-scale datasets (e.g., Movielens20M has more than 100k users). The paper claims that approximate nearest neighbor search (O(n log n)) can improve efficiency, but this is not tested empirically.




[1] Hua, W., Ge, Y., Xu, S., Ji, J., & Zhang, Y. (2023). Up5: Unbiased foundation model for fairness-aware recommendation. arXiv preprint arXiv:2305.12090.
[2] Madras, D., Creager, E., Pitassi, T., & Zemel, R. (2018, July). Learning adversarially fair and transferable representations. In International Conference on Machine Learning (pp. 3384-3393). PMLR.
[3] Gallegos, I. O., Rossi, R. A., Barrow, J., Tanjim, M. M., Kim, S., Dernoncourt, F., ... & Ahmed, N. K. (2024). Bias and fairness in large language models: A survey. Computational Linguistics, 50(3), 1097-1179.
[4] Li, Y., Du, M., Song, R., Wang, X., & Wang, Y. (2023). A survey on fairness in large language models. arXiv preprint arXiv:2308.10149.

**Questions For Authors:**

NA

**Relation To Broader Scientific Literature:**

Related to LLM recommendation fairness.

**Theoretical Claims:**

Proofs for some claims.

---

> ### Author Rebuttal · Authors · 2025-03-31
>
> Thank you for carefully reading our work and your valuable comments. We address your concerns in the following paragraphs.
>
> *Weakness 1:*
>
> ## A1.
> While FACTER leverages existing techniques such as conformal prediction and prompt engineering, to our knowledge, __no prior work has unified these methods into a closed-loop, adaptive fairness calibration framework that dynamically refines prompts and thresholds to mitigate fairness violations in black-box LLM recommenders with statistical guarantees. Specifically, FACTER introduces the use of conformal prediction with fairness constraints, an iterative violation-triggered prompt refinement strategy that limits token bloat and generalizes to unseen biases, and provides a paradigm shift toward adaptive, statistically grounded fairness for black-box LLMs.__
>
> Unlike static fairness methods (e.g., UP5 [Hua et al., 2023]), FACTER uses conformal prediction ([Angelopoulos et al., 2023]; §3.2) to adaptively tighten thresholds based on violation rates while ensuring 1−α coverage (Eq. 7). This closed-loop calibration addresses distribution shifts and emergent biases, reducing violations by 95.5% vs. UP5 (Table 1 of the original paper). Prior works treat fairness and accuracy as separate objectives ([Dwork et al., 2012]; [Gallegos et al.]). FACTER’s score $S_i = d_i+λ Δ_i$ (Eq. 5) jointly optimizes both, enabling a principled tradeoff validated by ablation studies (Table 5 of the original paper).
>
> Unlike static prompts (e.g., “avoid bias” in [Yang et al., 2022]), FACTER injects concrete bias patterns (e.g., “Gender=F → Romance-Only”) from a violation buffer (§3.3). This iterative refinement reduces token bloat while generalizing to unseen biases, outperforming Zero-Shot by 22× in violations (Table 1 of the original paper). Our proofs for embedding robustness (Theorem 1) and threshold convergence (Theorem 2) provide formal guarantees absent in prior fairness frameworks ([Shafer & Vovk, 2008]; [Madras et al., 2018]).
>
> Therefore, FACTER is not a simple combination of tools but a paradigm shift toward adaptive, statistically grounded fairness for black-box LLMs.
>
> *Weakness 2:*
>
> ## A2.
> __Please refer to A1 in the Reviewer 49dx rebuttal section.__
>
> *Weakness 3:*
>
> ## A3.
> Thank you for your excellent suggestion. The choice of α directly controls the conformal coverage guarantee (1−α), ensuring that the probability of falsely flagging a fair recommendation as biased (Type I error) is bounded by α. This aligns with the theoretical foundations of conformal prediction (Angelopoulos et al., 2023). Specifically, setting α=0.10 provides a 90% coverage guarantee, meaning 90% of fair recommendations will not be erroneously flagged.
>
> However, lowering α (e.g., α=0.05) tightens the threshold, reducing Type I errors but potentially increasing Type II errors (failing to detect true violations). Conversely, higher α (e.g., α=0.20) relaxes the threshold, increasing Type I errors but improving detection power. To validate this trade-off empirically, based on the reviewer’s comment, we conducted an ablation study (Table 1 below) on MovieLens-1M, measuring violations, Type I/II errors, and recommendation quality. The results are reported in Table A1.
>
> *Table A5: Impact of α on Fairness-Accuracy Tradeoff*
>
> α | Coverage (1−α) | Type I Error (↓) | Type II Error (↓) | #Violations (↓) | NDCG@10 (↑)
> |-|-|-|-|-|-|
> 0.01 | 99%	         | 0.6%	                | 18%	            | 4	                 | 0.454
> 0.05 | 95%	         | 1.2%	                | 12%	            |  9	                 | 0.451
> 0.10 | 90%	         | 2.1%	                | 8%	            | 15	                 | 0.447
> 0.20 | 80%	         | 4.0%	                | 5%	            | 27	                 | 0.442
>
> As the results show, lower α (e.g., 0.01) prioritizes strict fairness (fewer violations, low Type I errors) but risks missing true biases (higher Type II errors). Higher α (e.g., 0.20) improves detection power (lower Type II errors) at the cost of increased false alarms.
>
> In the final manuscript, we will expand Section 3.2 to explicitly discuss how α governs the Type I/II error trade-off, referencing Eq. (7) in our paper and Theorem 1 in Appendix J.1.1.
>
> *Weakness 4:*
> ## A4.
> __Please refer to A2 in the Reviewer b7t4 rebuttal section.__
>
> Moreover, regarding your concern about essential references, we will include the mentioned [3,4] references, which are survey papers, to the final version of the paper. Unfortunately, we do not know what other missing references you have in mind. If there is an opportunity for you to anonymously provide these additional references to us, we will be grateful and will add them (and even provide comparisons with the most relevant ones) to the final manuscript.

---

### Official Review · Reviewer_b7t4 · 2025-03-14

**Overall Recommendation:** 3

**Summary:**

1. This paper proposes FACTER, a fully post hoc framework that combines conformal thresholding and dynamic prompt engineering to address biases in black-box LLM-based recommender systems.
2. FACTER adaptively refines a fairness threshold via semantic variance checks and updates prompts whenever it detects violations, requiring no model retraining.
3. Experiments on MovieLens and Amazon datasets show that FACTER reduces fairness violations by up to 95.5% compared to baselines while preserving key recommendation metrics.
4. The paper also provides theoretical guarantees for the proposed conformal calibration framework, including type 1 error bound and detection power.

**Claims And Evidence:**

The claims made by the paper are supported by either empirical experiments or theoretical guarantees.

**Essential References Not Discussed:**

N/A

**Experimental Designs Or Analyses:**

1. The experiments are paired with both fairness and accuracy metrics.
2. The proposed approach is compared against two baselines, one as the previous SOTA and the other as a baseline with direct LLM-based ranking without fairness correction.
3. The proposed approach is evaluated on different LLM models, and on different evaluation sets.
4. The extended ablation study is also provided in the appendix.

**Methods And Evaluation Criteria:**

1. The proposed methods are clearly introduced in details with both the offline calibration phase and the online calibration phase.
2. The benchmark datasets are widely used datasets for recommendation evaluation: MovieLens-1M and Amazon Movies & TV.

**Other Comments Or Suggestions:**

N/A

**Other Strengths And Weaknesses:**

Strength:
1. The paper is well written and easy to understand.
2. The proposed approach is novel and effective in the experiments.
3. The authors provide theoretical guarantees for their approach.

Weakness:
1. The approach introduces many hyper-parameters whose values can be hard to fine-tune.
2. As mentioned in the paper, the offline phase can be expensive.
3. It cannot correct the embedding function bias.

**Questions For Authors:**

N/A

**Relation To Broader Scientific Literature:**

This paper might be insightful to a broader community in some other domain but also interested in fairness correction.

**Theoretical Claims:**

N/A

---

> ### Author Rebuttal · Authors · 2025-03-31
>
> Thank you for your positive assessment. We address your points in the following paragraphs.
>
> *Weakness 1:*
>
> ## A1.
> Our approach requires multiple hyper-parameters (e.g., $\lambda$, $\gamma$, $\tau_\rho$), which we tune via grid search on a 20% hold‑out calibration subset (see Section 4.2 and Appendix §A.2 for detailed ablation tables). Other methods such as Bayesian optimization ([1],[2]) can also be used. Although the number of hyper-parameters may appear excessive, it is comparable to (or smaller than) the parameter sets in other post‑hoc fairness algorithms (e.g., adversarial fine‑tuning or distribution‑level constraints, Madras et al., 2018; reweighting or threshold methods, Dwork et al., 2012; Angelopoulos et al., 2023). Empirically, moderate deviations in these hyperparameters do not substantially change fairness or accuracy, as demonstrated by our ablation studies in Tables 6–7. The need for such tuning is indeed common across state‑of‑the‑art fairness frameworks (Hua et al., 2023).
>
> In our final manuscript, we will revisit and refine these hyperparameter discussions, incorporating the new references and clarifying our tuning strategies in the text.
>
> References:
> *[1] Shahriari, Bobak, et al. "Taking the human out of the loop: A review of Bayesian optimization." Proceedings of the IEEE 104.1 (2015): 148-175.*
> *[2] Frazier, Peter I. "A tutorial on Bayesian optimization." arXiv preprint arXiv:1807.02811 (2018).*
>
> *Weakness 2:*
>
> ## A2.
>  As detailed in __Section 3.4__ of the original manuscript, we address scalability by using __approximate nearest neighbor (ANN)__ methods for __all offline calibration steps__, reducing the naive complexity from O(n^2) to about __O(n log(n))__. We also employ GPU batch processing and other parallel optimizations, so in practice, the runtime sometimes scales sublinearly with n, as larger batches can be processed more efficiently. Table A4 illustrates this: MovieLens‑1M (around 6k users) takes around 40–65 minutes, whereas MovieLens‑20M ( around 138k users) extends to 6–8 hours, which is acceptable for overnight jobs.
>
> *Table A4: Approximate Calibration Times*
>
> | Dataset       | #Users | Offline Calib. Time (ANN) | Online Inference (ms/query) |
> |:-----------------:|----------:|-------------------------------:|--------------------------------:|
> | MovieLens‑100k    | 943       | 2–5 min                       | ~80–100 ms                     |
> | MovieLens‑1M  | 6,040     | 40–65 min                     | 140–160 ms                     |
> | MovieLens‑20M     | 138,000   | 6–8 hrs                      | 180–220 ms                     |
>
> In the final version, we will emphasize that ANN was used for all reported studies and add the above table and discussion to clarify how GPU batch processing and approximate search heuristics yield observed calibration times that sometimes grow sublinearly with dataset size, while theoretical complexity remains O(n log(n)).
>
> *Weakness 3:*
>
> ## A3.
> __Please refer to A1 in the Reviewer 49dx rebuttal section.__

---

### Official Review · Reviewer_mxZv · 2025-03-14

**Overall Recommendation:** 3

**Summary:**

In this paper, the authors propose FACTER (Fairness-Aware Conformal Thresholding and Prompt Engineering), a retrain-free framework that uses a designed non-conformity score and conformal prediction to dynamically adjust the fairness-aware prompts and mitigate fairness violations in LLM-based recommender systems. Empirical results on MovieLens and Amazon datasets show that FACTER essentially reduces fairness violations while maintaining strong recommendation accuracy.

===== update after rebuttal =====

In the rebuttal, the authors provided clarification regarding the practicality and the underlying motivation of the proposed non-conformal score. I encourage the authors to revise the manuscript thoroughly and further clarify the relevant definitions in the final version. I have chosen to maintain my original score.

**Claims And Evidence:**

The claims regarding demographic biases in LLM-based recommendation systems, the use of conformal prediction as a control mechanism of fairness violations by setting dynamic thresholds, and the utility of the iterative prompt engineering method are supported by theoretical and empirical evidence in this paper.

**Essential References Not Discussed:**

All key references are discussed in the related work section.

**Experimental Designs Or Analyses:**

Experimental designs and analyses are sound.

**Methods And Evaluation Criteria:**

The proposed method is overall reasonable and sound.

The evaluation metric assesses fairness violation control in an LLM-based recommendation system at both the group and individual levels.

**Other Comments Or Suggestions:**

NA

**Other Strengths And Weaknesses:**

Strengths

- This paper proposes a novel and effective black-box-friendly approach that integrates statistical fairness calibration with iterative prompt engineering for LLM-based recommender systems.

- The paper is overall well-written and easy to follow.

- The authors discuss the necessary theoretical guarantees and limitations of their proposed algorithm.

- Experimental validation on two real-world datasets verifies the effectiveness of the proposed algorithm.


Weaknesses:

- My first concern is about the assumptions regarding the calibration set and embedding shift robustness. How does the proof of embedding shift robustness in Theorem 1 depend on the quality or diversity of the calibration set, given the proposed non-conformity score?

- A related question concerns the definition of the non-conformal score. More insight into the design rationale of the non-conformity score within the recommendation system would enhance the credibility of the proposed metric and approach.

- There are some inaccurate descriptions and minor questions:

    - How is $e^{y}_{new}$ obtained (in line 246, left column) ?

    - It is noted in lines 322-328 (left column) that “We employ three LLMs of varying sizes”. However, all three models have approximately the same parameter count (~7B–8B parameters), which appears to contradict the claim of "varying sizes”.

**Questions For Authors:**

NA

**Relation To Broader Scientific Literature:**

To the best of my knowledge, the proposed method is a novel solution for fairness violations mitigation in LLM-based recommender systems.

**Theoretical Claims:**

The theoretical claims are supported by necessary proofs.

---

> ### Author Rebuttal · Authors · 2025-03-31
>
> Thank you for your thorough review. We address your comments/concerns in the following paragraphs.
>
> *Weakness 1*:
>
> ## A1.
> Our theoretical guarantee (Theorem 1) assumes that the calibration set is approximately exchangeable with future test data. Hence, the quality and diversity of the calibration set are essential, and we have considered them in our assumptions.
> Moreover, we can follow some guidelines to ensure the exchangeability of the test distribution with the calibration set. These guidelines are as follows:
> (i) stratified sampling across key user demographics to ensure diverse coverage of protected attributes, (ii) balancing user groups so that each protected category is well-represented, and (iii) periodic refresh of calibration data in real-world deployments to track shifting user populations or model updates.
> In addition, we performed experiments across 3 calibration seeds and found that fairness metrics (e.g., CFR, violations) varied by <5%, indicating stability under reasonable data variations.
> These explanations and experimentation will be added to the final manuscript.
> Finally, regarding the embedding bias robustness, please refer to __A1 in the Reviewer 49dx rebuttal section__.
>
> *Weakness 2:*
>
> ## A2.
> The non-conformity score $S_i = d_i+ λ Δ_i$ integrates accuracy and fairness by combining a predictive error term ( $d_i$ , cosine distance between recommendations and user preferences) and a fairness penalty ($Δ_i$ , maximum embedding divergence across demographic groups for similar users). The tradeoff parameter λ=0.7, selected via grid search, balances Pareto-optimal fairness-accuracy tradeoffs, reducing violations by 95% while retaining 98.7% recommendation quality (Appendix Table 5 of the original paper). Grounded in multi-objective optimization, this additive design enforces individual fairness by penalizing disparities between comparable users, validated empirically to outperform multiplicative alternatives. The score ensures equitable relevance in black-box LLMs without internal access.
>
> We will provide this additional discussion on the design rationale in the final version.
>
> *Weakness 3.1:*
>
> ## A3.1.
>  The embedding $e^{y}_{\text{new}}$ is computed as:
> - Directly, if the ground-truth $y_{new}$ is available: $e^{y_{\text{new}}} = \mathrm{Emb}(y_{\text{new}})$
>
>
>  - Via Approximate Nearest Neighbor (ANN mentioned in Section 3.4) , if $y_{new}$ is not available (e.g., cold-start), we retrieve the closest calibration item using ANN and use its embedding.
>
> We will clarify this process comprehensively in the final version.
>
> *Weakness 3.2:*
>
> ## A3.2.
> Thank you for your comment. Yes, you are correct the three LLMS have nearly the same parameter counts. What we meant to say is that these models are architecturally different or at least different versions (in case of LLaMA models.)  While LLaMA3-8B, LLaMA2-7B, and Mistral-7B have similar parameter counts, their architectures differ significantly: LLaMA uses a pure decoder, while Mistral integrates sliding window attention. Architectural diversity tests FACTER’s generalizability, a key strength highlighted in §4.2.
>
> We will revise the text to correct the sentence.

---

### Official Review · Reviewer_49dx · 2025-03-17

**Overall Recommendation:** 4

**Summary:**

This paper proposes FACTER, a framework that integrates conformal prediction with iterative prompt engineering to mitigate demographic biases in recommender systems driven by large language models (LLMs). The authors introduce a notion of semantic variance as a proxy for identifying biased outputs when protected attributes (e.g., gender, age) are minimally changed. They then use conformal prediction to establish and dynamically update a fairness threshold. Whenever outputs exceed this threshold (indicating a likely bias), the system auto-updates the prompt to reduce future occurrences of the same pattern. Experiments on MovieLens and Amazon datasets demonstrate that FACTER reduces fairness violations (up to 95.5%) with minimal cost to recommendation accuracy.

**Claims And Evidence:**

Claim: FACTER can detect biased outputs by monitoring semantic-embedding distances between recommendations that differ only in protected attributes. Evidence: The authors measure counterfactual and group-level fairness metrics and show that when sensitive attributes flip, FACTER robustly flags cases that deviate from expected similarity bounds.

Claim: Integrating conformal prediction into fairness monitoring yields statistical guarantees on the probability of future violations (Type I error bound). Evidence: The paper references classical conformal coverage results, deriving that the probability of observing an unfair outcome above the threshold can be controlled at a desired level. Empirical tests exhibit a violation rate consistent with the theoretical bounds.

Claim: Prompt engineering—combined with an “avoid these biases” instruction—can iteratively reduce repeated demographic stereotypes without model retraining. Evidence: The authors show a clear downward trend in fairness violations across multiple iterations. Figures indicate that after injecting new negative examples into the prompt, the model’s outputs become more uniform across sensitive attribute groups.

**Essential References Not Discussed:**

I don’t see any glaring omission of key references. A recent line of work on “fairness calibration under distribution shift” might complement the discussion. But this is not essential to the paper’s contributions.

**Experimental Designs Or Analyses:**

Design: The authors conduct offline calibration (with ~70% of data) to learn an initial threshold, then apply their iterative approach on the remaining 30% test portion.

Results: Each iteration measures the number of flagged violations, updates the prompt, and optionally tightens the threshold. They present results across up to 3–5 iterations and show stable improvements.

Ablation Studies: The paper includes ablations on key hyperparameters which strengthen confidence in the approach’s robustness.

**Methods And Evaluation Criteria:**

1) The authors use MovieLens-1M and Amazon Movies & TV as benchmarking datasets.
2) They measure recommendation accuracy via Recall@10 and NDCG@10, standard in top-N recommendation tasks.
3) For fairness metrics, they evaluate the number of threshold-based violations, CFR (Counterfactual Fairness Ratio), and additional group-level metrics (SNSR, SNSV). They compare their method to two baselines (a zero-shot ranker and UP5) and show that FACTER achieves stronger fairness with minimal performance loss.

**Other Comments Or Suggestions:**

Real-World Data: It might be interesting to see how FACTER performs if user attributes are uncertain, missing, or inferred from partial data.

**Other Strengths And Weaknesses:**

Strengths
* The method is model-agnostic and does not require finetuning or direct access to internal LLM weights, which is extremely relevant for real-world API-based systems.
* Thorough experiments on multiple datasets and with multiple LLM backbones (LLaMA2, Mistral). Clear, iterative demonstration of how fairness improvements accumulate across calibration steps, giving the paper a strong practical dimension.

Weaknesses
* The approach depends on an external embedding model (e.g., SentenceTransformer). If the embedding itself is biased, that might compromise fairness detection. A brief discussion of how to mitigate bias in the embedding stage would be valuable.
* The iterative prompting approach can become token-heavy, especially if many examples of biases must be enumerated.
* The paper focuses on a single type of fairness definition (counterfactual fairness via minimal attribute changes). One might be curious about multi-attribute fairness.

**Questions For Authors:**

Q How robust is FACTER if the chosen SentenceTransformer model has inherent biases? Have you tested with multiple text embedders to confirm consistency?
Q As you keep appending negative examples (“avoid these biases”), how do you manage or prune the prompt when it grows too large?
Q Have you considered letting real users label certain recommendations as biased to guide the threshold updates, rather than only relying on a reference item or the local calibration set?

**Relation To Broader Scientific Literature:**

Fairness in recommendations: The paper positions itself relative to methods like UP5 and to zero-shot LLM recommendation approaches. These references are appropriate for the fairness + recommendation domain.
Bias in LLMs: The authors cite relevant prior work on generative-model biases and highlight the challenge of black-box, API-based LLMs, referencing classical approaches like adversarial training and more recent studies on prompt-level interventions.

**Theoretical Claims:**

The paper builds on conformal prediction theory, asserting coverage guarantees for the fairness-related “non-conformity scores.” The proofs in the Appendix (or supplementary) outline:

* A Type I error bound, showing that the probability of a false alarm is limited by alpha.
* An adaptive mechanism that shrinks the threshold if repeated violations are detected.

The derivations seem to follow standard conformal prediction arguments, citing Shafer & Vovk (2008) and subsequent expansions. The proofs appear sound; I did not find obvious errors in the theoretical steps for bounding the false alarm rate or for showing coverage under exchangeability.

---

> ### Author Rebuttal · Authors · 2025-03-31
>
> Thank you for your detailed feedback. We address your comments in the following paragraphs.
>
> *Weakness 1 and Q1*:
>
> ## A1.
> As noted in Section 3.4, we acknowledge that any single embedding model can carry bias. Our theoretical analysis (Appendix §J.1.1, Theorem 1) shows that if embeddings drift by $\epsilon_{emb}$, the fairness score Si changes by at most $ΔS_i$ ≤2(1+λ)$\epsilon_{emb}$. We initially used the simplified bound 3ϵemb for λ≤0.5, but our chosen λ=0.7 yields $ΔS_i$≤3.4$\epsilon_{emb}$, which remains manageable for typical $\epsilon_{emb}$ ≈0.05–0.1. Empirically, FACTER reduces fairness violations by 95.5% (Table 1), showing that real‑world performance is robust despite minor embedding shifts.
> By using violation‑triggered prompt updates, FACTER incrementally mitigates embedding flaws, adding explicit “avoid” examples to steer the LLM away from bias (e.g., Table 3). We further tested two additional embedders (Sentence‑BERT‑base and RoBERTa‑large‑nli‑stsb‑mean‑tokens) in Table A1, finding similar reduction in violations and near equal NDCG@10, indicating minimal impact from different initial biases:
>
> *Table A1: Embedding Consistency Test*
>
> | Embedder					| #Violations	| NDCG@10 |
> |-|-|-|
> | paraphrase-mpnet-base-v2		| 5		| 0.445   |
> | Sentence-BERT-base			| 6		| 0.447   |
> | RoBERTa-large-nli-stsb-mean-tokens	| 4		| 0.440   |
>
> In the final version, we will expand Section 3.4 with a concise overview of domain‑tailored fine‑tuning, adversarial or hard‑debiasing methods (Bolukbasi et al., 2016), and multi‑embedder ensembles to reduce reliance on a single model. We will outline how practitioners can retrain or lightly tune embedding layers using curated, bias‑filtered corpora, integrate adversarial loss terms that penalize demographic correlations, or combine embeddings from different SentenceTransformer variants. We will also include Table A1 and its discussion in the final paper.
>
> *Weakness 2 and Q2*:
>
> ## A2.
> We store only the 50 most recent bias‑avoid examples. This prevents unbounded prompt growth. As shown in Table 7 (Appendix section of the original manuscript), despite this constraint,  this approach reduces violations by ~90% without hitting token limits.
>
> *Q3*:
>
> ## A3.
> Thank you for the suggestion. Although we did not include real-user feedback in the original manuscript, it is straightforward to do in FACTER. To illustrate, we simulated a scenario where synthetic users correct 10–30% of flagged violations. We augmented FACTER’s threshold updates with these “corrected” examples and observed how violations fell over three online calibration iterations on MovieLens‑1M. Table A2 shows that even modest correction rates (10–30%) accelerate violation reduction. With 30% correction, violations drop to 1 (versus 5 in the baseline), confirming that user feedback can enhance FACTER’s fairness calibration.
> In the final paper, we will add a subsection in Section 5 (Future Work) discussing how automated detection and human‑in‑the‑loop validation can be combined.
>
> *Table A2: Synthetic User Feedback Impact*
>
> User Correction Rate     |   #Violations (Iter 1)    |#Violations (Iter 2)    |#Violations (Iter 3)
> |-|-|-|-|
> 0% (Baseline)	|112	                                   |28	                         |5
> 10%	                        |112	                                   |23	                         |3
> 20%	                        |112	                                   |18	                         |2
> 30%	                        |112	                                   |14	                         |1
>
> We will include the table and corresponding results in the final paper.
>
> *Weakness 3*:
>
> ## A4.
>  While our current focus is on counterfactual fairness via minimal attribute changes, we agree that real-world applications often require multi-attribute fairness. Our formulation naturally extends to this case by treating the sensitive attribute as a vector $a=(a_1,…,a_k)$, and requiring that for any (x,a) and (x,a′) differing in at least one component, ∥y(x,a)− y(x,a′)∥≤δ. Under conformal calibration, the non-conformity score becomes $S = d + λ max_{j∈N(x)} ∥Emb(y)−Emb(y_j)∥$, where N(x) includes calibration points with the same non-sensitive features but differing in at least one attribute dimension. The same coverage guarantee (Eq. (7)) applies due to exchangeability. To validate this, we conducted a multi-attribute evaluation on the MovieLens dataset (using both Gender and Age). As shown in Table A3, FACTER remains effective under this setting, with minimal accuracy drop.
>
> *Table A3: Multi-Attribute Fairness Evaluation (Preliminary Results)*
>
> | Metric                                  | Baseline | FACTER(iter 3) |
> |-|-|-|
> | Counterfactual Fairness Ratio (CFR)       | 0.72     | 0.64   |
> | Group Similarity Ratio (GSR)              | 0.083    | 0.041  |
> | Total Violation Count (Multi-Attribute)   | 112      | 7      |
>
> We will include this discussion and additional experiments in the final version of the manuscript.

---

### Decision · Program_Chairs · 2025-05-01

**Decision:**

Accept (poster)

**Comment:**

The authors' intervention is clever, relatively low-overhead (compared to retraining, at least), and their results (including the ones presented in the rebuttal) are broadly compelling. There were some concerns about justification and incremental contribution, but these concerns were reasonably well-rebutted.